# Crosstalk between Depression and Breast Cancer via Hepatic Epoxide Metabolism: A Central Comorbidity Mechanism

**DOI:** 10.3390/molecules27217269

**Published:** 2022-10-31

**Authors:** Zhen Ye, Kumar Ganesan, Mingquan Wu, Yu Hu, Yingqi She, Qianqian Tian, Qiaobo Ye, Jianping Chen

**Affiliations:** 1School of Basic Medical Sciences, Chengdu University of Traditional Chinese Medicine, Chengdu 610075, China; 2School of Chinese Medicine, LKS Faculty of Medicine, The University of Hong Kong, Hong Kong; 3Department of Pharmacy, Sichuan Orthopedic Hospital, Chengdu 610093, China; 4Department of Social Science, The Hang Seng University, Hong Kong; 5Department of Breast Surgery, Guangdong Provincial Hospital of Traditional Chinese Medicine, Guangzhou 510405, China; 6Shenzhen Institute of Research and Innovation, The University of Hong Kong, Shenzhen 518057, China

**Keywords:** depression, epoxide metabolism, comorbidity, tumor microenvironment, breast cancer

## Abstract

Breast cancer (BC) is a serious global challenge, and depression is one of the risk factors and comorbidities of BC. Recently, the research on the comorbidity of BC and depression has focused on the dysfunction of the hypothalamic–pituitary–adrenal axis and the persistent stimulation of the inflammatory response. However, the further mechanisms for comorbidity remain unclear. Epoxide metabolism has been shown to have a regulatory function in the comorbid mechanism with scattered reports. Hence, this article reviews the role of epoxide metabolism in depression and BC. The comprehensive review discloses the imbalance in epoxide metabolism and its downstream effect shared by BC and depression, including overexpression of inflammation, upregulation of toxic diols, and disturbed lipid metabolism. These downstream effects are mainly involved in the construction of the breast malignancy microenvironment through liver regulation. This finding provides new clues on the mechanism of BC and depression comorbidity, suggesting in particular a potential relationship between the liver and BC, and provides potential evidence of comorbidity for subsequent studies on the pathological mechanism.

## 1. Introduction

Cancer is one of the malignant diseases with the highest mortality in the world, and its incidence continues to grow rapidly [1,2]. The leading cause of cancer-related mortality among the female population is breast cancer (BC) [3]. Since 2020, BC has been the major cause of cancer incidence worldwide, accounting for 11.7% of all cancer cases [3,4]. BC is also the fifth highest cause of cancer deaths around the world [3]. In addition, BC is an obstacle to improving life expectancy in each country, causing a heavy economic burden and health and social challenges globally [3]. BC is a highly heterogeneous disease whose development is associated with genetic, dietary, and environmental factors [5]. Various types of BC can be broadly classified as hormone receptor status (estrogen receptor, ER, or progesterone receptor, PR), human epidermal growth factor receptor status (HER2), and triple-negative status (TNBC) [5,6,7,8]. The current mainstream treatment options include conventional chemotherapy, monoclonal antibodies, and coupled systemic administration [5]. Although increased levels of diagnosis of BC over the years have led to increased survival rates, the side effects of treatment, the impact of stress, and the unsatisfactory quality of survival have still attracted public concern [9]. BC has a high rate of physical and mental comorbidity, mainly due to chronic stress [10]. Depression, as a vital risk factor and comorbidity of BC, has plagued women with BC for decades. In the absence of the precise management of individuals, families, and professional domains, female BC patients are placed under mental stress, which eventually brings a heavier physical burden. Studies have illustrated that depression is an independent predictor of higher frequency hospitalization, longer hospitalization, lower quality of life, and lower treatment compliance [11]. Depression has also been demonstrated to be an important predictor for the diagnosis of advanced BC patients, and the suicide rate of BC patients has highly correlated with clinical symptoms of depression phenotypes [12]. It was reported that comorbid depression is associated with poor prognosis and increased mortality in cancer patients [13]. A study demonstrated that the prevalence of depression in BC patients is 15% during and after medical cancer treatment. The treatment of anxiety and depression are associated with decreased neurocognitive function and reduced hippocampal volume following chemotherapy [9,14,15,16,17]. What is more, in the context of the current global prevalence of infectious diseases, BC patients are prone to emotional disturbances and cognitive dysfunction due to the impact of work and employment [18]. A meta-analysis demonstrates that negative emotions significantly increase the risk for the incidence of BC [19]. Thus, the comorbidity of BC and depression is an inescapable biomedical problem.

Hitherto, most studies addressing the comorbidity of BC and depression have focused on four aspects: inflammation and oxidative/nitrosative stress, reduced immune monitoring, abnormal activation of the autonomic nervous system, and the hypothalamic–pituitary renal axis (HPA) [20]. In fact, the imbalance of peripheral dopamine (DA) and kynurenine (KYN) are proposed to positively predict depression in BC patients [21]. Moreover, the persistent activation of the HPA and sympathetic nervous system is believed to promote BC growth. Unfortunately, due to the dispersion of information, the bridging mechanism between depression and BC is still unclear since the etiology and final effect of the comorbidity have only been partly discussed. Epoxide metabolism is an important metabolic process that mediates inflammation, tumor, and immune surveillance, which mainly occurs in the liver, kidney, and blood vessels [22].

Epoxide metabolism is noted to play a significant regulatory role in BC. Soluble epoxide hydrolase (sEH) is an essential intermediate enzyme in epoxide metabolism and has a vital effect on the pathogenesis of depression and BC [22,23]. Several studies have indicated that upregulation of sEH is closely related to neurological disorders [24]. A decrease in sEH level is also found in BC tissues, whereas an increase in sEH level inhibited BC proliferation. Other scholars have shown that sEH can promote BC cell proliferation by hydrolyzing toxic epoxides, which is inconsistent with previous studies [25,26]. Therefore, sEH-mediated epoxide metabolism might be a crucial area for investigation and one of the critical comorbid mechanisms of BC and depression. However, the presented evidence is controversial. According to the study, epoxide metabolism mainly occurs in the liver, and sEH may have different effects on different subtypes of BC. Furthermore, epoxide metabolism is involved in mediating immune responses and regulating lipid homeostasis in the tumor microenvironment (TME) [27]. Researchers have demonstrated that the levels of plasma interleukin 6 (IL-6) in patients with BC and depression are higher and are also regulated by sEH [28,29]. The epoxide metabolism mediated by sEH might be related to a deeper mechanism, which is the key point of the controversy.

According to previous studies, the relationship between depression and BC, as well as depression and the liver, has been partially demonstrated. However, the potential link between BC and the liver has attracted little attention. Basically, this paper focuses on the role of hepatic epoxide metabolism as an essential bridging mechanism between BC and depression, providing evidence for the pathogenesis of depression and BC comorbidity as well as clarifying the variabilities in sEH effects across BC subtypes. The literature review identifies the liver for the first time as the contributing organ to one of the comorbidities of depression and BC partially, providing a reference for further in-depth analysis of depression and BC comorbidity.

## 2. Depression Is an Important Risk Factor and Comorbidity of BC

Depression, one of the reported risk factors for cancer, is known as a comorbidity of BC. Researchers have found that BC survivors experience a high rate of depression, and the incidence of depression during and after treatment is 15% [9,30]. Current research has underlined the neurohormonal signaling system as the major shared mechanism of BC and depression. The sympathetic nervous system (SNS) and HPA are two stress responses that affect the nervous system and contribute to BC development [31]. When depression occurs, chronic stressors activate the HPA axis, resulting in adrenaline and catecholamines release. Following the HPA axis activation, adrenaline activates BC-adrenergic receptors, accumulates myeloid-derived suppressor cells (MDSCs), and promotes BC development [32]. Cortisol secreted by the adrenal cortex promotes BC cell development by activating the glucocorticoid receptor (GR) signaling pathway, serum/glucocorticoid-regulated kinase 1 (SGK1), and mitogen-activated protein kinase phosphatase 1 (MKP1)/dual-specific phosphatase 1 (DUSP1) [33]. At the same time, cortisol leads to a reduction in tumor immunosurveillance by suppressing immune function with decreased natural killer (NK) cell activity and T cell proliferation [34] (Figure 1).

Further, depression is associated with BC partly due to the increase in macrophage activity induced by depressive phenotypes. The M1 macrophages are an important factor in inflammation in patients with severe neurological disorders [35]. Research on major depressive disorder found elevated levels of circulating cytokines in peripheral blood mononuclear cells (PBMCs), as well as increased levels of NF-kB in PBMCs [36]. Adipocytes and BC tumor cells release chemokines (e.g., C-C motif chemokine ligand 2 (CCL2), C-C motif chemokine ligand 5 (CCL-5), or colony-stimulating factor (CSF-1)) to promote the migration of monocytes and macrophages into the BC microenvironment [37,38]. These macrophages contain M1 and M2 phenotypes, while the M1 macrophages are always transformed into M2 within the BC microenvironment, and so are the monocytes [39,40,41,42]. Adipocytes in the breast stroma are an important source of interleukin 10 (IL-10), which also contributes to the polarization of macrophages to the M2 phenotype in BC [43,44,45]. Clinical studies have also indicated that IL-10 is an independent factor in poor prognosis in TNBC, ER-negative, or PR-negative cases [46,47].

Chronic systemic inflammation induced by prolonged stress in depression has clearly been shown to be an initiating factor in carcinogenesis [48]. IL-6, one of the proinflammatory cytokines, is a signaling promoter and pathological product of depression [49]. Studies have shown that high levels of IL-6 are related to the chronic course of depression, and the severity of depression in patients with high expression of IL-6 is increased as well. The research results of elderly patients with depression show higher levels of IL-6 than in healthy elderly people [50]. Likewise, IL-6 also plays an instrumental procancer role in BC. Clinical evidence indicates that IL-6 induction is associated with a poor prognosis for a patient with BC, with plasma IL-6 levels showing a positive correlation with pathological grade. A preclinical study derived that the IL-6/IL-6R/gp130 pathway promotes the growth and metastasis of BC, while inhibiting the pathway is not conducive to the development of BC. Therefore, IL-6 may contribute to BC and depression in comorbid states. Additionally, TNF-α is a pathogenic cytokine in depression. One study found that the levels of proinflammatory cytokines TNF-α and IL-6 in patients with major depression increased significantly [51]. Meanwhile, anti-TNF-α drugs are found to be antidepressants [52]. The dual effect of TNF-α on BC is discussed as well [53]. The immune response further suggests that chronic inflammation is an important basis for depression and BC comorbidity. The fact of the hormone regulation and cytokine effect have been widely mentioned, but the intermediate stage of the pathogenesis of the comorbidity is still unclear.

## 3. Specific Changes in Epoxide Metabolism in Individuals with Depression

In the past few decades, studies on depression have mainly focused on brain dysfunction. The brain is rich in polyunsaturated fatty acids (PUFAs), including omega-6 polyunsaturated fatty acids (ω-6 PUFAs) and omega-3 polyunsaturated fatty acids (ω-3 PUFAs), which are the precursors of arachidonic acid (AA), linoleic acid (LA), eicosapentaenoic acid (EPA), and docosahexaenoic acid (DHA). Cytochrome P450 (CYP450), one kind of enzyme existing in microsomes and mitochondria, converts AA and LA to epoxyeicosatrienoic acids (EETs) and epoxyoctadecamonoenoic acids (EpOMEs), while eicosapentaenoic acids (EPAs) and docosahexaenoic acids (DHAs) are also converted to epoxyeicosatetraenoic acids (EpETEs) and epoxydocosapentaenoic acids (EpDPAs) [54,55,56]. sEH is a hydrolase mainly existing in hepatocytes, which is also distributed in the kidney and vascular endothelium [22]. The epoxides that are converted via CYP450 can be degraded by sEH into dihydroxyeicosatetraenoic acids (DiHETEs), DiHOME, DiHETrE, DiHTPA, which maintain low biological activity [22]. sEH has been demonstrated to increase neurological inflammation, which is closely related to EET degradation. In patients with anorexia, plasma sEH activity is also significantly increased. [57]. The occurrence of plasma sEH activity builds a basis for depression and BC. Depression was once believed to be caused by the hippocampus, the most important regulatory hub of emotion. Studies have exposed that sEH inhibitors can replenish epoxy lipids and reduce neuroinflammation and amyloid pathology [57,58].

Meanwhile, sEH inhibitors alleviate depression by altering the BDNF–trkB signaling pathway in the hippocampus [59,60]. The positive effects of sEH inhibitors on depression are reported to be related to anti-inflammatory and repair-promoting EETs produced by arachidonic acid (AA) through cyclooxygenase (COX) metabolism, lipoxygenase (LOX) metabolism, and CYP450 enzymes [58,60,61,62]. Therefore, sEH is considered as a potential target for the treatment of psychiatric disorders such as depression and bipolar disorder, whose mechanisms are related to metabolic, inflammatory, and vascular factors [55,57,58,63] (Table 1 and Table 2).

## 4. Different Status of sEH Mediates Epoxide Metabolism in BC

sEH-mediated epoxide metabolism not only operates in depression but also in the formation of the BC microenvironment. However, the effect of sEH on BC is controversial. On the one hand, the sEH causes the decrease in EETs, a promoting factor on BC [68]. Meanwhile, 14, 15-EET can promote angiogenesis in BC tissues and upregulate vascular endothelial growth factor (VEGF) [69]. A clinical study revealed that 14, 15-EET induced integrin αvβ3 expression and FAK/PI3K/AKT activation, which strengthen stromal production and cisplatin resistance in BC cells (MCF-7 and MDA-MB-231) in vivo [70].

On the other hand, a cytotoxic leukocyte metabolite can be degraded to diol by sEH, indicating that diols can stimulate the proliferation of MCF-7 cells (ER+PR+HER2-) [24,71,72,73]. More seriously, inflammatory breast cancer (IBC) is considered to be a rapidly developing and metastasizing type of BC. The degradation of EETs leads to the generation of a proinflammatory microenvironment, which in turn becomes a risk factor for IBC. Hence, it appears that sEH has a further promotional influence on BC. Researchers have found that BCs with different receptor phenotypes have different levels of sEH, implying that epoxide metabolism involved in sEH will have different effects on different subtypes of BC, or even maybe the opposite [7,25]. Several pieces of evidence for the sEH influence on different phenotypes of BC have been identified via omics-based pathway analyses [7]. The study demonstrated that sEH expression is highly positive in HER2+ (75%), ER+PR+ (68%), and triple-positive BC (TPBC) (67%), but the weakest expression appeared in TNBC (46%). Meanwhile, the sEH inhibitory effects are significantly lower in HER2+ (33% strongly positive for sEH) BC, especially compared with TNBC. TNBC maintains the lowest sEH expression, while CYP450, including CYP2C8, 2C9, 2J2, and CYP3A4, exhibits strong expression in TNBC, which is known to promote the conversion of AA, LA, DHA, and EPA into EETs, EpOMEs, EpETEs, and EpDPAs. The EETs derived from these CYP450 have been confirmed to promote invasion and metastasis in TNBC [68]. Even though TNBC causes sensitive reactions to CYP450 and EETs, a different subtype of TNBC still shows the opposite phenomenon [7]. TNBC has been reported to be divided into several subtypes, including the mesenchymal-like subtype and basal-like subtype. [74,75,76,77]. Depletion of endogenous CYP450 attenuates the metastatic phenotype of mesenchymal-like TNBC cells [7]. Conversely, CYP2C19 depletion or any compound inhibitor treatment has no significant effect on the migratory and invasive potential of basal-like TNBC cells, and the inhibition of sEH failed to induce significant changes in total EET levels in any of the basal-like TNBC cell lines [7]. The metastatic and invasive capacity of basal-like TNBC is independent of CYP450. CYP450-mediated EET metabolisms have a stronger correlation with mesenchymal-like TNBC.

The expression or activity of targeted CYP450 has a more effective influence in reducing the metastatic burden of the mesenchymal-like TNBC subtype [7]. Meanwhile, HER2+ BC is found unaffected by either of the EETs. On the contrary, another study attributed the tamoxifen resistance in MDA-MB-361 (ER−/PR−/HER2+) to overexpression of CYP3A4, partly by enhancing 11,12-EET biosynthesis [78], while it is mentioned in the study by Maria Karmella Apaya that MCF-7 could not be affected by any EET [7]. Contradictory evidence reveals the existence of deeper complicated network mechanisms. Of note, sEH maintains a higher expression in hormone receptor-positive BC with the possible production of leukotoxins, further suggesting that sEH may be a key factor in the comorbidity of depression and hormone receptor-positive BC. sEH might promote the growth of BC by degrading epoxides and generating specific toxic diols, particularly in hormone receptor-positive BC.

## 5. sEH-Mediated Epoxide Metabolism Is Involved in the TME of BC

The occurrence of cancer is not a simple aggregation of a single cancer cell. The TME is an inevitable accompaniment of malignant tumors, and BC is no exception [31,79]. Lipid metabolism is substantiated to create a breeding ground for BC cells [80,81]. Basically, obesity has been confirmed as an established risk factor for BC [37,82,83]. As a major contributor to obesity, adipocytes are the main component of BC stroma. EETs can promote atherosclerosis metabolism to maintain adipocytes in order to ensure the energy supply for BC. This might be the crucial mechanism of the EET promotion for BC [70]. Free fatty acids released by adipocytes can be leveraged by BC cells to produce ATP [84]. Therefore, lipid metabolism is highly activated in BC tissue. Adipose tissue is enriched with several proinflammatory immune cell types such as M1 macrophages, T helper 1 (Th1), CD8 T cells, neutrophils, mast cells, and B cells [82]. When adipose tissue inflammation occurs, these inflammatory cells are activated, releasing interleukin 1β (IL-1β), TNF-α, IL-6, and monocyte chemoattractant protein 1 (MCP1) to induce insulin and leptin resistance [82]. At the same time, adipocytes become hyperplastic and hypertrophic, which is favorable for nutrient uptake by BC cells [83]. Studies have shown that peripheral BC cells of breast tumors can ingest fatty acid, while those inside the tumor are unable to do so [81,85,86]. Therefore, the peripheral BC cells are exposed to adipose tissue and proliferate, while the interior has no proliferation activity, which indicates that BC cells cannot proliferate greatly without adipocytes [79,81]. In BC cells, EETs promote the nuclear translocation of FABP4 and FABP5, activate TF lipid metabolic pathways, and upregulate the nuclear accumulation of PPARγ and SREBP-2. The nuclear accumulation of PPARγ and SREBP-2 can activate oncogenic cascades, including PPARγ binding to and degrading Nur77, and promote energy uptake from adipocytes [87].

Importantly, lipids from adipocytes surrounding breast tumors are reported through synergistic adipose triglyceride (TG)-adipose-dependent (ATGL-dependent) lipolysis and uncoupled FAO [80,88]. It is worth mentioning that sEH has a negative effect on cholesterol metabolism, which is related to the involvement of EETs. Trans-4-{4-[3-(4-trifluoromethoxyphenyl)-ureido] cyclohexyloxy} benzoic acid (t-TUCB), an inhibitor of sEH, has been demonstrated to increase hepatic 17, 18-epoxyeicosatetraenoic acid (17,18-EEQ) and 19, 20-epoxydocosapentaenoic acid (19, 20-EDP) levels and enhance the tissue inflammation and lipid peroxidation observed in ω-3 PUFA-dependent reduction [88]. t-TUCB also has a stronger antilipid effect on obese mice. Notably, t-TUCB biased macrophages toward an anti-inflammatory M2 phenotype and expanded the volume of interscapular brown adipose tissue. Although brown fat alleviates lipid inflammation, BC cells appear to benefit from other pathways, including the macrophage M2 phenotype [89]. Thus, the local and systemic effects induced by abnormal lipid metabolism create a tumor-friendly TME, which further promotes the initiation, invasion, and metastasis of BC. The inhibition of sEH increases the positive effect of ω-3 PUFAs on regulating lipid metabolism (Figure 2 and Figure 3).

Cholesterol homeostasis in breast adipose tissue is the breeding ground for BC proliferation, metastasis, and angiogenesis pathways (e.g., Tips or VEGF) [80]. BC-associated adipocytes are connected with the early and late stages of BC. Early BC invades adipocytes in white fat tissue in exchange for cytokines and signaling molecules, whereas BC cells induce the lipolysis and adipocyte phenotype as cancer-associated adipocytes (CADC), an abnormal adipocyte with low lipid droplet accompanied by abnormal expression of adipokine secretion, including leptin, lipocalin, IL-6, CCL2, and CCL5 [90,91,92,93,94,95]. In addition, the infiltration of EETs induces the transformation from white to brown adipose tissue, accompanied by reduced inflammatory expression in adipocytes, which seems to be detrimental to BC [96]. Other studies have suggested that BC is highly associated with brown fat deposition [97,98,99]. Since brown adipocytes are characterized by a large number of mitochondria and high vascular density, BC may benefit from brown adiposity through angiogenesis-related mechanisms [100]. This suggests that BC may benefit from white adipose tissue, inflammatory adipose tissue, and brown adipose tissue at the same time [99,100]. Meanwhile, CADC reorganizes the actin cytoskeleton, increases fibroblast-like biomarkers including fibroblast-activating protein a (FAP), chondroitin sulfate proteoglycan, and smooth muscle actin (a-SMA), and transforms into a fibroblast-like morphology [81].

## 6. Depression-Associated sEH Promotes Liver Dysfunction and Breast Cancer

As mentioned above, with the increase in depression exploration, extracerebral pathological alterations have been paid more and more attention. As the main organ of lipid metabolism, especially the metabolism of epoxide, the liver has been proposed to have an unexpected connection with brain functions [101,102,103,104,105]. The pathogenesis of Alzheimer’s disease (AD) is believed to be closely related to depression [106,107]. In particular, major depression and AD share biological processes and pathways, including 77 disease-related genes and 102 pathways [108]. Amyloid β (Aβ) is one of the vital shared pathogenic targets [109,110,111,112]. AD has been pointed out to be significantly associated with liver dysfunction [113]. The liver and kidney are the main extracerebral organs for the clearance of circulating Aβ [113]. Low-density lipoprotein receptor 1 (LRP-1), which is mediated through the liver, has been shown to regulate multiple tight junction proteins in the blood–brain barrier endothelium [114]. Studies also showed that those central mechanisms are involved in the neurological and endocrine systems of the brain leading to neuroendocrine regulation of CYP450 gene expression. Such mechanisms have been shown to involve the dopaminergic, noradrenergic, and 5-hydroxytryptaminergic systems of the brain with hypothalamic endocrine centers, among which repetitive restraint stress (RS) increases hepatic CYP2D1/2 activity in a stress-specific manner, while the main effectors of the stress system, glucocorticoids and epinephrine, are highly induced by CYP3A1/2 [66,115,116,117,118,119]. Epinephrine also induces the expression of CYP2C11 and CYP2D1/2. Studies have shown that human hepatocyte microsomes are the primary site of systemic CYP450, sEH-mediated epoxide metabolism [120]. This discovery further supports the importance of hepatocyte epoxide metabolism in depression–BC comorbidity.

Researchers also disclosed that psychiatric disorders induce liver dysfunction [113]. A study found that sEH in the livers of chronically stressed experimental animals increased specifically without its appearance in other organs and caused nonspecific changes in LOX and COX signaling pathways [64]. Furthermore, the specific knockdown of the hepatic sEH gene Ephx2 suppressed the expression of the depression-like phenotype. Moreover, the important hydrolase is predominantly expressed in hepatocytes [64]. The evidence suggests that hepatic sEH is one of the main causative factors of psychiatric disorders including depression. Additionally, sEH has been raised as the critical molecule in the brain–liver axis, with a positive correlation between sEH protein in the parietal cortex and sEH protein in the liver [67]. Thus, the downstream effects of hepatic epoxide metabolism, including inflammation, liver dysfunction, and lipid metabolisms, may be components of contributors to BC.

The effects of depression on the liver are known to be reflected in inflammation, oxidative damage, and reduced immune surveillance [101,105,121]. From the perspective of molecular mechanisms, the upregulation of hepatic sEH is one of the pivotal upstream causes of liver damage, liver fibrosis, and hepatitis [101,122,123,124]. The inhibition of hepatic sEH significantly reduces endoplasmic reticulum stress in hepatocytes and maintains low expression of prostaglandins and triglycerides, thereby reducing high-fat-diet-induced inflammation [89]. A notable observation is that overexpression of hepatic sEH increases liver triglyceride levels and hepatic inflammatory response [101,105].

Surprisingly, the induced expression of sEH in the liver only occurred in a long-term rather than short-term high-fat diet. Furthermore, sEH inhibition attenuates the high-fat-diet-induced plasma levels of proinflammatory cytokine increase and the adipocytic cytokine mRNA upregulation. It is thus clear that depression promotes specific upregulation of hepatic sEH, which is a potential inflammatory injury factor in the liver [54,89]. What is more, overexpression of sEH in the liver directly affects the balance of epoxide metabolism. The pathological changes in the liver are crucial to the disruption of the internal environmental homeostasis, which provides a potentially favorable environment for the development of BC.

Another noteworthy point is that ω-6 PUFAs and ω-3 PUFAs, as the precursors of DHA, EPA, AA, and LA, are clearly involved in the pathogenesis of depression [65]. Studies have shown that ω-3 PUFAs exert anti-inflammatory effects in the brain by regulating microglia function to maintain homeostasis, which improve fatty acids for the pathogen. At the same time, ω-6 PUFAs have been considered as promoting factors of inflammation, while the ability of dietary LA to increase the levels of inflammatory markers is influenced to some degree by the level of adiposity [65,125]. The ratio of ω-6 PUFAs/ω-3 PUFAs is considered to affect the balance of lipid metabolism [125,126]. A Mediterranean diet with high ω-3 PUFA levels has been mentioned to alleviate depressive symptoms and even decrease the prevalence of malignancies such as breast, lung, prostate, and colorectal cancers [127,128]. In addition, ω-3 PUFAs have been found to attenuate microglia-induced inflammation by inhibiting the HMGB1/TLR4/NF-κB pathway [129].

It has also been shown that an unhealthy diet can lead to obesity, which is highly correlated with a chronic inflammatory environment and depression. A high ratio of ω-6 PUFAs/ω-3 PUFAs has been demonstrated to be an unhealthy dietary pattern [130,131]. Western diets, another popular diet construction, show a high ω-6 PUFAs/ω-3 PUFAs ratio of 15/1 to 16.7/1 [130,132]. Western diets have been shown to induce hyperthrombotic and proinflammatory states [131,133,134,135,136]. ω-6 PUFAs and ω-3 PUFAs can be converted into EPA, DHA, AA, and LA after ingestion, which can be further converted into EETs, EpOMEs, EpETEs, and EpDPAs. This shift suggests that the Western diet promotes an imbalance in lipid metabolism, which leads to a range of proinflammatory and carcinogenic effects [22,23,137]. In addition, the dietary structure of high ω-6 PUFAs/ω-3 PUFAs is conducive to the growth and development of BC. Studies have discovered that LA is the most abundant polyunsaturated fatty acid in the Western diet [125,126], where the consumption of butter, corn oil, the rice plant (*Oryza Sativa* L.), and soybeans leads to significantly increased hepatic LA consumption and promotes LA metabolism via CYP450 (mainly by CYP2J2, CYP2C8, and CYP2C9) to produce 9, 10-EpOME (leukotoxin) and 12, 13-EpOME (isoleukotoxin) [138,139,140,141,142,143]. Another important fact is that stress is reported to induce the hepatic PXR expression, which is followed by induced hepatic CYP3A and CYP2C expression [116]. Therefore, the response of high dietary ω-6 PUFAs to depression results in high sEH expression promoting the production of 9, 10-DiHOME (leukotoxin diol) and 12, 13-DiHOME (isoleukotoxin diol) [69,117]. In addition, a high ω-6 PUFAs/ω-3 PUFAs diet may increase breast cancer risk [144,145,146,147]. Moreover, it has been shown that LA promotes the proliferation of MDA-MB-231 breast cancer cells by activating the EGFR/PI3K/Akt signaling pathway [148,149]. Meanwhile, AA has been found to have a similar effect as well [149]. It is worth noting that excessive LA intake has been implicated in the development of obesity and liver dysfunction [97,150,151,152,153,154,155]. This effect is fatal to BC patients, especially BC with obesity. Basically, hepatic sEH-mediated epoxide metabolism is an important mechanism by which the liver regulates the comorbidity of depression and BC.

Under the stimulation of chronic depression, chronic stress leads to the continuous increase in inflammatory cytokines leading to chronic persistent inflammation. Although the hypothalamic–adrenal axis is one of the important regulatory pathways of chronic inflammation induced by chronic stress, sEH-mediated epoxide metabolism is also an important regulatory pathway [118]. This is because EETs degraded by sEH are also important contributors to the creation of a proinflammatory endotrophic environment. Fatty liver has been identified as consistent with increased neuro-proinflammatory cytokines and amyloid β deposition in the brain of mice induced by a high-fat diet [156]. Thus, abnormal lipid metabolism and hepatic inflammation mediated by disturbances in hepatic epoxide metabolism are directly related to neuroinflammation and pathological alterations in amyloid β in the brain. Importantly, sEH upregulation increases EET degradation and promotes cytokine expression. A study illustrated that dual inhibitors of sEH and COX-2 improved hepatic fibrosis and portal hypertension and downregulated IL-6 levels, suggesting that IL-6 plays a driving role in hepatitis [157]. The prevalence of nonalcoholic fatty liver diseases (NAFLD) in patients with breast cancer is significantly higher than in healthy controls, while hepatic sEH is a key enzyme for NAFLD [158]. Moreover, breast cancer patients with NAFLD showed poorer prognosis in terms of recurrence [158,159].

In addition, as mentioned earlier, depression-mediated neuroinflammation is associated with IL-6, and there is also a positive relationship between this interleukin and the construction of TME in BC [160]. Therefore, IL-6-related pathways may be one of the downstream pathological developmental pathways of epoxide metabolic disorders. Currently, several studies are aimed at blocking the IL-6 receptor and its downstream signaling molecules to develop BC-related therapeutic regimens [29,161]. Generally, systematic chronic inflammation is a beneficial environment for BC development, which is tightly connected with adipose issues (especially cancer-associated adipocytes) [95,161,162,163,164]. This review points out that another potential target, blocking hepatic sEH rather than breast tissue sEH, might be a suppressor in partially comorbid BC and depression populations (Table 3).

## 7. Prospective Studies

Chronic stress stimulates neuropathological changes and induces depression. The pathological effects of depression include an imbalance in epoxide metabolism and inflammatory response, which leads to the degradation of EETs with anti-inflammatory and repair effects. Chronic stress-induced psychiatric disorders lead to the development of hepatic dysfunction and the specific upregulation of hepatic sEH. The series of hepatic abnormalities, after persistent progression, directly develop hepatic inflammation and lipid metabolism disorders. Although sEH is found to be downregulated in malignant breast tissues and its upregulation inhibits the proliferation of BC, due to the high heterogeneity of BC cells and the complexity of BC pathogenesis, the effect of sEH is not isolated, one-sided, or guarded.

In this review, we conclude that sEH downregulation has the least effect on hormone receptor-positive BC, while sEH downregulation has the greatest effect on TNBC promotion. Moreover, the obvious effect of sEH is related to the involvement of EETs in the regulation of adipocyte homeostasis in breast tissue and thus lipid acquisition by peripheral cells of tumor tissue. When EETs are degraded, the tumor tissue might become “starved”. Interestingly, the lack of EETs leads to an inflammatory phenotype of adipocyte appearance, accompanied by an increase in lipid droplets and even lipid efflux because of adipocyte fragmentation. These pathological changes might be precisely the facilitating factors for BC. Moreover, BC is also beneficial for different types of adipose tissue. Unfortunately, the mechanism underlying the interaction between brown adipose tissue and BC remains unclear. Furthermore, when BC is accompanied by ω-6 PUFA excessive intake (especially high-LA diets), sEH is associated with a large release of leukotoxic diol, which stimulates BC cell proliferation and promotes BC development. Of note, the BC-protective mechanism awaits further study in vivo. Most importantly, although the development of BC shows a negative correlation with sEH, it needs to be clarified that the sEH-mediated effects are still different from organs. Therefore, the role of hepatic epoxide metabolic imbalance and inflammatory response in promoting BC cannot be ignored, especially for IBC. However, it is important to note that the production of 9, 10-EpHOME, the precursors of 9, 10-DiHOME, is in parallel with the production of EETs. The dominant secretion of these two epoxy fatty acids that occur under different conditions remains unclear and needs to be further explored. Moreover, based on the dual regulation of PUFAs, the appropriate strategy of regulating PUFA intake is still a meaningful effort.

Depression, by inducing hepatic dysfunction and hepatic epoxide metabolism disorders, allows the development of systemic chronic inflammation, which has been demonstrated to be a prominent component of BC. Inflammatory response recruits BC tissue through the joint action of BC cells and microenvironmental factors, polarizes macrophages and monocytes into M2 macrophages, participates in the construction and expansion of TME, and promotes the development of tumors. In addition, hepatic epoxide metabolism is crucial for the homeostasis of lipid metabolism. Based on the integrated effect of depression-related pathological processes, abnormalities in epoxide metabolism, and inflammatory responses, hepatic lipid metabolism disorders lead to the production and accumulation of adipocytes. The pathological state creates a high-fat internal environment, which happens to be the perfect condition for BC survival and growth.

The hepatic sEH-mediated epoxide mechanism may be a bridging mechanism from depressive phenotypes to BC, which directly contribute to the TME of the breast. Crucially, the present review puts forward a novel point: the status of sEH-mediated epoxide metabolism in participating in depression and BC comorbidity may not be confined to BC cells or even locally to breast tumors. This comorbidity mechanism varies with subtypes. Moreover, this bridging mechanism may be significantly associated with the TME, and therefore the influence of exogenous molecules provided by the TME on BC cells cannot be ignored. Notably, sEH acts on the liver producing an indirect procarcinogenic effect, whereas sEH acts directly on BC cells as a protective effect. It should be noted that chronic stress is the main cause of hepatic sEH-specific upregulation in this review. The specific upregulation of hepatic sEH occurs not only in those with diagnosed depression but also in those with a positive phenotype of interest (Figure 4).

Even in patients with undiagnosed depression, the prolonged stimulation of a depressive mood still has a high potential to evoke the liver-specific mechanism. In the contemporary era with many adverse factors such as social stress and work competition, the potential mechanism of bridging the depressive phenotype to BC appears important. The liver-specific upregulation of sEH induced by the depressive phenotype may be one of the critical mechanisms that support depression exacerbating BC. The hepatic sEH-mediated BC-promoting mechanism cannot be directly equated with the sEH expression of BC cells, among which the byproducts may be the major risk factors. The situation suggests that further investigation of sEH requires special attention to the dyadic relationship, i.e., the same molecule of BC can not only produce subtype differences but also derived effects due to its distribution in peripheral organs. Nevertheless, this comprehensive review is limited by the literature, and whether depression-induced hepatic sEH upregulation can promote BC development directly still needs more investigation. All in all, the dialectical cognition towards the bidirectional regulation of epoxide metabolism may exemplify an innovative mechanism for further clinical and preclinical studies and provide a reference for the target discovery and development of new treatment protocols for BC and depression comorbidities in the future (Figure 5).

## 8. Conclusions

This comprehensive review described the crosstalk between depression and BC through hepatic epoxide metabolism. The comorbidity of depression and BC is associated with all-cause BC mortality and organ metastasis. The review identified the possibility that hepatic epoxide metabolism could be a comorbidity and etiology of BC progression. This comorbidity is caused due to alterations in lipid metabolism, overexpression of inflammation, and upregulation of toxic diols. This verdict provides new evidence on the mechanism of comorbidity between BC and depression, particularly signifying a potential connection between the liver generating epoxide and BC, thus providing potential evidence of comorbidity for subsequent investigations on the pathological mechanism. More comprehensively designed preclinical and clinical investigations are greatly required to validate these potential mechanisms.

## Figures and Tables

**Figure 1 molecules-27-07269-f001:**
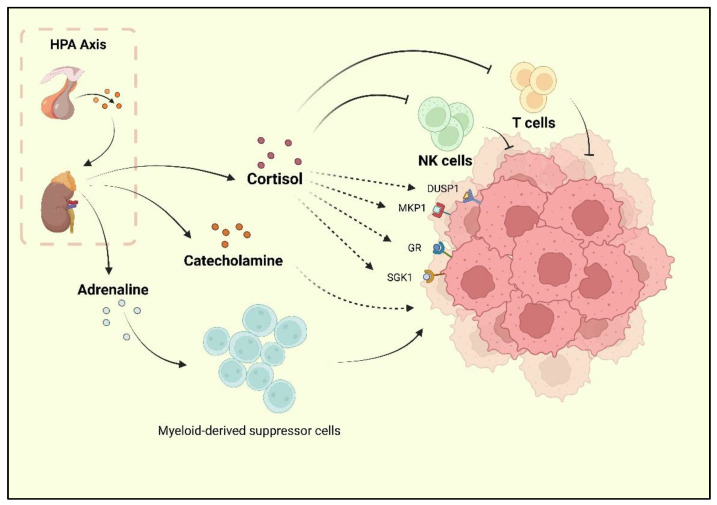
The HPA axis activated by stress is reported to promote cortisol, adrenaline, and catecholamine release. Cortisol could inhibit the activity of NK cells and T cells, as well as promote the activation of GR, MKP1, SGK1, and DUSP1, which are the positive factors for breast cancer progression. Catecholamine also plays a promoting role for BC. Moreover, adrenaline could upregulate the bioactivity of MDSCs, which indirectly promote the progression of BC.

**Figure 2 molecules-27-07269-f002:**
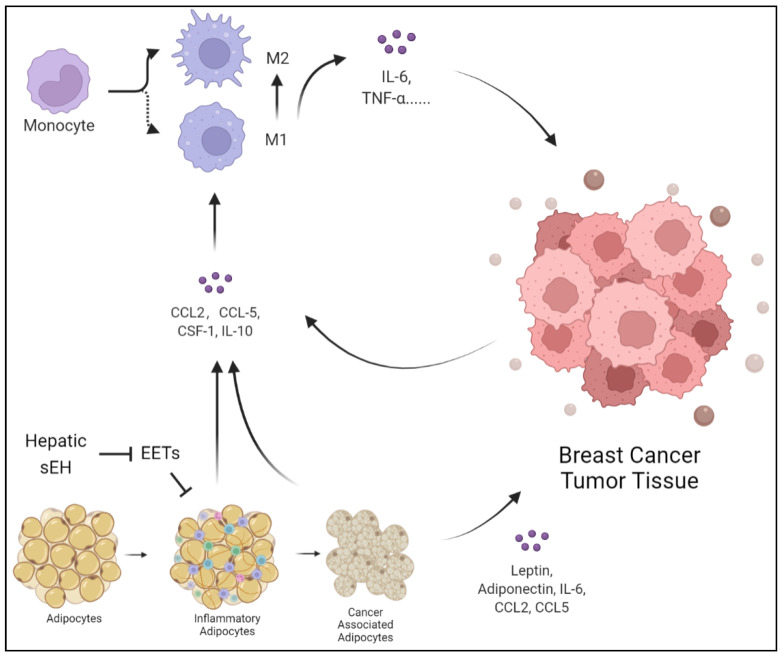
Adipocytes can be transformed into inflammatory cells under hepatic injury, releasing CCL2, CCL5, CSF-1, and IL-10. The products of inflammatory adipocytes can promote monocytes and macrophages to release IL-6 and TNF-α, polarize into M2 macrophages, and thus promote tumor growth. At the same time, the hepatic sEH overexpression degrades EETs, which inhibit the suppressive effect on inflammatory adipocytes. Additionally, BC cells induce adipocytes to transform into cancer-associated adipocytes and highly express leptin, adiponectin, IL-6, CCL2, and CCL5.

**Figure 3 molecules-27-07269-f003:**
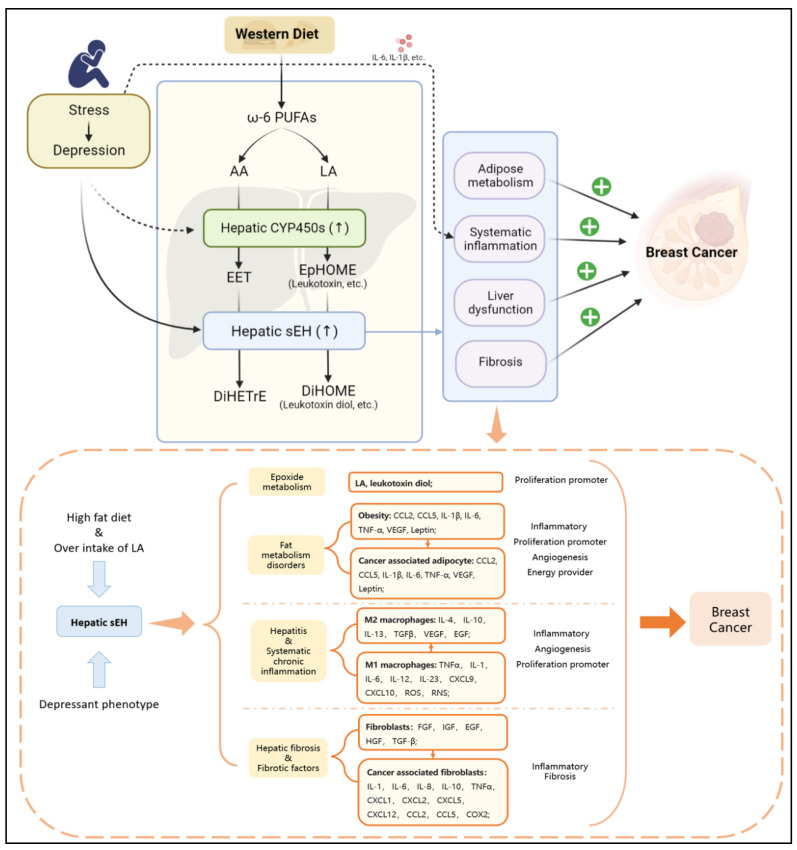
The depressant phenotype can stimulate the specific upregulation of hepatic sEH, while a high-fat diet (especially overintake of LA) can cause an excessive level of LA in the body. The secondary effects related to BC have four types of risk factors: epoxide metabolism, fat metabolism disorders, hepatitis and systematic chronic inflammation, hepatic fibrosis and fibrotic factors. The risk factors can promote BC via angiogenesis, inflammation, fibrosis in TME, and progression of proliferation.

**Figure 4 molecules-27-07269-f004:**
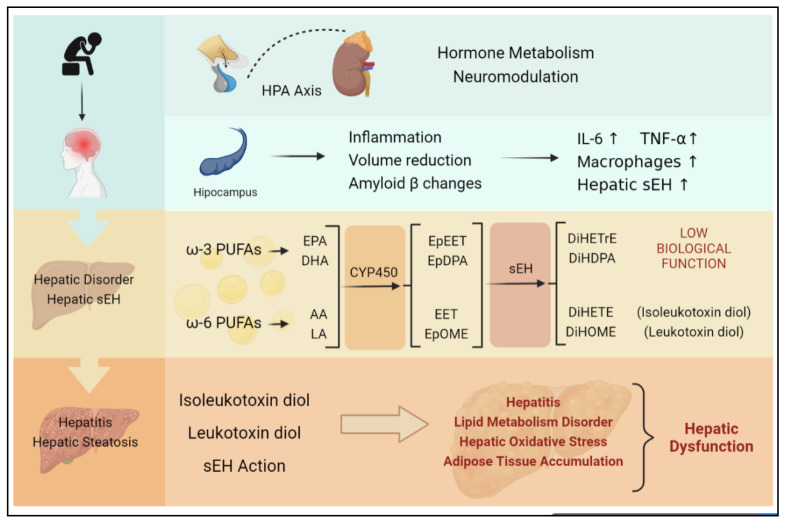
Chronic stress is largely induced by continuous negative emotion, which is likely to develop into depression. Both chronic stress and depression can activate the HPA axis, stimulate the hippocampus, and induce inflammation, as well as hepatic sEH upregulation specifically. The upregulation of sEH in the liver can lead to an imbalance in epoxide metabolism and excessive production of DiHETrE, DiHDPA, DiHOME (containing leukotoxin diol), and DiHETE (containing isoleukotoxin diol). The degrading of EET suppresses the anti-inflammation effect. At the same time, toxic diols lead to hepatic dysfunction, including hepatitis, lipid metabolism disorder, hepatic oxidative stress, and adipose tissue accumulation, which directly promotes the occurrence of nonalcoholic hepatitis.

**Figure 5 molecules-27-07269-f005:**
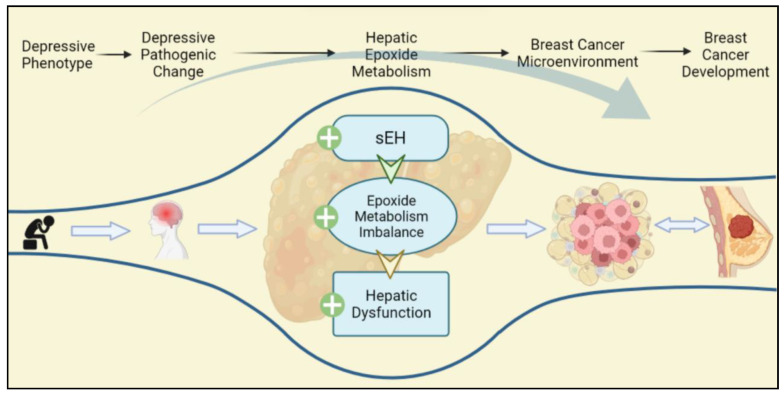
Depressive phenotypes are the prestage of depression, which can also lead to the depressive pathogenic change progressing. Depressive phenotypes specifically upregulate hepatic sEH, which contribute to hepatic dysfunction, epoxide metabolism imbalance, and lipid metabolism disorder. The hepatic pathogenic change promotes the TEM of BC, thereby strengthening the development of BC.

**Table 1 molecules-27-07269-t001:** Association between hepatic sEH and depression based on the preclinical studies.

S. No	Model/Animals Used	Route of Administration	Outcome of the Study	Key Findings	References
1	CMS C57BL/6J mice (in vivo).	NA.	RNA-seq: 1. Hepatic *Ephx2* mRNA (↑); 2. ARA metabolic pathway significantly changed.	Hepatic sEH activity was selectively altered followed by CMS. The reductions in sucrose preference and coat score that were induced by CMS were both rescued by imipramine and fluoxetine (chronic antidepressant treatment).	[64]
2	CMS C57BL/6J mice (in vivo).	NA.	Western blot: 1. Hepatic monomeric sEH (↑); 2. Hepatic oligomeric sEH (↑); 3. sEH in heart, kidney, spleen, lung, muscle, stomach, PFC, or hippocampus (-).		
3	CMS C57BL/6J mice (in vivo).	NA.	Sucrose preference (↓); coat score (↓).		
4	CMS C57BL/6J mice (in vivo).	Tail vein injections/AAV-*Ephx2*-shRNAs, AAV-NC (control)/15 days.	CMS mice injected with AAV-NC virus: 1. Anhedonia (↓), sucrose preference (↓), coat score (↓); 2. The mRNA and protein levels: hepatic sEH (↑). CMS mice injected with AAV-*Ephx2*-shRNAs: 1. Anhedonia (↑), sucrose preference (↑), coat score (↑); 2. The mRNA and protein levels: hepatic sEH (↓).	Hepatic sEH may play an important role in depression.	[64]
5	CMS C57BL/6J mice (in vivo).	NA.	The 1-week intervention of CMS did not affect corticosterone secretion. Corticosterone administration did not affect sEH expression in primary cultured hepatocytes.	Hepatic sEH was independent of corticosterone-related mechanisms.	[64]
6	CMS cKO mice/CMS WT mice (in vivo).	NA.	CMS-induced WT mice: sucrose preference (↓); coat score (↓); CMS-induced cKO mice: sucrose preference (↑); coat score (↑).	Hepatic deletion of *Ephx2* ameliorated CMS-induced depressive phenotypes and related inflammatory response. Deletion of the hepatic *Ephx2* gene reversed the CMS-induced reduction in synaptic transmission and plasticity. Deletion of the hepatic *Ephx2* gene cannot affect the levels of IL-6, MCP-1, and Iba1.	[64]
7	CMS cKO mice/CMS-induced WT mice (in vivo).	NA.	CMS-induced WT mice: 1. CAMK IIα in the hippocampus (↓); CAMKIIβ in the hippocampus (↓); Arc in the hippocampus (↓); 2. Neurotrophic factors mRNA and proteins in the mPFC and the hippocampus: BDNF (↓); GDNF (↓); GFAP (↓); EAAT1 (↓); EAAT2 (↓). CMS-induced cKO mice: The hepatic deletion of *Ephx2* reversed the CMS-induced downregulation.		
8	CMS cKO mice (in vivo).	NA.	The levels in the brains of cKO mice: macrophage/microglia activity markers Iba1 mRNA (↓); cyclin-dependent kinase 11b mRNA (↓); macrophage inflammatory protein 1α mRNA (↓).		
9	CMS C57BL/6J mice; CMS cKO mice (in vivo).	NA.	The level in the mPFC and the hippocampus: IL-6 (↑); MCP-1 (↑); Iba1 (↑).		
10	C57BL/6J mice (in vivo).	Tail vein injections/pEf1a-*Ephx2* plasmid DNA, pEf1a-eYFP (control)/detection after 3 days.	Immunohistochemistry: eYFP fluorescence was positive only in liver sections. The mRNA levels on the third day after injection: Hepatic sEH mRNA (↑); hepatic sEH oligomer levels (↑); hepatic sEH monomer levels (↑). Behavior analysis: sucrose preference (↓); immobility with forced swim test (↑). Protein expression: GluN2A in the mPFC (↓); GluN2B in the mPFC (↓).	Overexpression of hepatic *Ephx2* gene-induced depressive phenotypes.	[64]
11	C57BL/6J mice (in vivo).	Tail vein injections/AAV-*Ephx2*, AAV-eYFP (control)/evaluation after 3 days.	The mRNA, monomer, and oligomer expression levels: hepatic sEH (↑). Behavior analysis: sucrose preference (↓); immobility with forced swim test (↑).		
12	cKO mice (in vivo).	NA.	Female cKO mice: plasma levels of 14, 15-EET (↑). Male cKO mice: plasma levels of 14, 15-EET (↑).	14, 15-EET production is specifically regulated by the liver. 14, 15-EET plasma levels are positively correlated with sucrose preference and coat scores. 14, 15-EET plasma levels are positively correlated with sucrose preference.	[64]
13	CMS mice (in vivo).	NA.	Plasma levels of 14, 15-EET (↓).		
14	CMS mice; cKO mice (in vivo).	Tail vein injection/AAV-*Ephx2*, pEf1a-*Ephx2*/evaluation after 4 weeks.	Female mice and the mice injected with AAV-Ephx2 or pEf1a-*Ephx2*: plasma levels of 14, 15-EET (↓). cKO mice: plasma levels of 14, 15-EET (-).		
15	cKO mice; WT littermates (in vivo).	i.c.v. infusions of 14,15-EEZE/3 weeks; a cannula was implanted in the right ventricle of cKO mice or WT littermates.	cKO mice: The antidepressant-like effect (↓) after 30 min infusions. Behavior analysis: sucrose preference (-); immobility with forced swim test (-).		
16	Neurons were isolated from the mPFC or the hippocampus of neonatal C57BL/6J mice (in vivo).	14, 15-EET/3 h.	The mRNA levels of synaptic proteins and neurotrophic factors (-).		
17	Neurons; astrocytes (in vitro).	Coculture.	Hippocampal neurons mRNA levels: Grin2a mRNA (↑), Grin2b mRNA (↑), Gria1 (↑), Gria2 (↑), Camk2a (↑), Camk2b (↑), Arc (↑), Bdnf (↑), Gdnf (↑), Ngf (↑), Vegfa (↑), Vegfb (↑).	Astrocytic EET signaling in the mPFC mediated the effects of hepatic *Ephx2* deletion. sEH is expressed primarily in astrocytes in the brain, and astrocyte EET can bind and regulate blood flow to neuronal activity. Astrocytes are the target cells for exogenous 14, 15-EET.	[64]
18	C57BL/6J mice (in vivo).	Infusing into mPFC or the hippocampus via a cannula; 14, 15-EET; evaluation after 30 min.	Infusion into mPFC: immobility (↓), locomotor activity (-). Infusion into the hippocampus: immobility (-), locomotor activity (-).		
19	cKO mice; WT littermates (in vivo).	Lenti-*hEPHX2* injection Lenti-EGFP (control).	Lenti-hEPHX2 injection in the mPFC: sEH (↑); CMS cKO mice injected with Lenti-EGFP: resilience (↑); Lenti-hEPHX2 injection in the mPFC of WT littermates: anhedonia (↑), sucrose preference (↓), and coat deterioration (↑). The antidepressant-like effect induced by the hepatic deletion of *Ephx2* was blocked by the overexpression of *hEPHX2* in the mPFC. The increased protein levels of GluA1, GluA2, and GluN2A in the mPFC of cKO mice were attenuated by the overexpression of *hEPHX2* in the mPFC.		
20	Male C57BL/6 mice (8–10 weeks old); model: FST (in vivo).	0.01, 0.1, 1 mg/kg TPPU (sEHI)/i.p./30 min.	0.01, 0.1 mg/kg TPPU and 1 mL imipramine reduce immobility time in mice during FST.	Acute administration of sEHI TPPU decreases depressant phenotypes. Intraperitoneal administration of TPPU for seven days also reverses the depressant phenotypes.	[59]
21	Male C57BL/6 mice (8–10 weeks old); model: NSF (in vivo).	0.01, 0.1, 1 mg/kg TPPU (sEHI)/i.p./30 min.	1. Feed latency (↓); 2. Improvement in antidepressant phenotypes and cell proliferation were inhibited by BDNF–trkB antagonist K252a; 3. Hippocampal BDNF expression (↑), cell proliferation in the dentate gyrus (↑).		
22	Female and male mice in which microglia expressed the green fluorescent protein (GFP; fms-EGFP or MacGreen mice) (in vivo).	A single intraperitoneal injection of LPS (1 mg/kg).	1. Neuroblast number (↓), whose effect was exacerbated by the ω-3 PUFA-deficient diet; 2. The ω-3 PUFA-deficient diet reduced the DG volume, AHN, microglia number, and surveilled volume; 3. The diet effect on most mature neuroblasts was exclusively significant in female mice.	Colocalization and multivariate analysis revealed an association between microglia and AHN, as well as the sexual dimorphic effect of diet. Female mice are more susceptible than males to the effect of dietary ω-6/ω-3 PUFA ratio on AHN and microglia.	[65]
23	MD Wistar rats; RS Wistar rats; MDR Wistar rats (in vivo).	NA.	Plasma corticosterone levels: RS rats (↑), MD rats (↑); CYP3A2 expression at the mRNA, apoprotein, and activity (6β-testosterone hydroxylation) level: RS rats (↑), MD rats (↑), MDR rats (↑); CYP3A1 expression at the mRNA, apoprotein, and activity (6β-testosterone hydroxylation) level: MD rats (↑), MDR rats (↑); CYP2C11 expression at the mRNA and activity level: MDR rats (↑); CYP2D1 expression at the mRNA, apoprotein, and activity (6β-testosterone hydroxylation) level: RS rats (↑); CYP2D1 expression at the mRNA: MDR rats (↑).	1. MD and RS regulate CYP expression. 2. The expression of CYP3A1 and CYP2C11 was increased in the liver of MD rats, whereas RS had no significant effect. 3. Hepatic CYP2D1/2 activity was increased by RS, whereas MD did not affect it.	[66]
24	Wistar rats (in vivo).	Phenylephrine hydrochloride, 2 mg/kg i.p., 1 × 4, α1-agonist.	CYP3A1 transcripts (↑), CYP3A2 expression (↑).		
25	Primary hepatocytes (in vitro).	Corticosterone (CORT; 1–25 μM).	CYP2C11 transcripts (↑), CYP2D2 transcripts (↑).		
26	Primary hepatocytes (in vitro).	Epinephrine (10 μM, 24 h).	CYP3A1 expression (↑), CYP3A2 expression (↑), CYP2C11 transcripts (↑).		

**Abbreviations:** Chronic mild stress (CMS); genome-wide RNA sequencing (RNA-seq); prefrontal cortex (PFC); forced swim test (FST); novelty-suppressed feeding test (NSF); soluble epoxide hydrolase (sEH) inhibitor (sEHI); 1-Trifluoromethoxyphenyl-3-(1-propionylpiperidin-4-yl)urea (TPPU); intraperitoneal injection (i.p.); tamoxifen (TAM); calcium-/calmodulin-dependent protein kinase (CAMK) IIα (encoded by Camk2a); CAMKIIβ (encoded by Camk2b); Arc (encoded by Arc); brain-derived neurotrophic factor (BDNF) (encoded by Bdnf); glial cell-derived neurotrophic factor (GDNF) (encoded by Gdnf); glial fibrillary acidic protein (GFAP) (encoded by Gfap); glial-specific excitatory amino acid transporters (EAAT1 and EAAT2) (encoded by Eaat1 and Eaat2); macrophage/microglia activity markers Iba1, cyclin-dependent kinase 11b (encoded by Cdk11b), and macrophage inflammatory protein 1α (encoded by Ccl3); interleukin (IL)-6; monocyte chemoattractant protein 1 (MCP-1) (encoded by Ccl2); ionized calcium-binding adaptor molecule (Iba1) (encoded by Iba1); 14,15-epoxyeicosa-5(Z)-enoic acid (14,15-EEZE), a selective EET antagonist; intracerebroventricular (i.c.v.); nonsignificant change (-); not acceptable (NA); significant increase (↑), *p* < 0.05; significant decrease (↓), *p* < 0.05; no significance (-); cKO: crossed albumin-CreERT2 mice with Ephx2loxp/loxp mice to generate a double-transgenic mouse model (Ephx2 was conditionally deleted in the livers of adult mice via TAM-inducible Cre recombination); maternal deprivation stress (MD); repeated restraint stress (RS); maternal deprivation stress (MDR); nonresponders in the maternal deprivation stress (MDNR); epoxyeicosatrienoic acid (EET); arachidonic acid (AA); linoleic acid (LA); dihydroergotoxine (DHET).

**Table 2 molecules-27-07269-t002:** Association between hepatic sEH and depression based on the clinical studies.

S. No	Samples	Outcome of the Study	Key Findings	References
1	Blood plasma from MDD patients.	ELISA: 14, 15-EET (↑); 5, 6-EET (-); 8, 9-EET (-); 11, 12-EET (-).	1. A change in the ratio of epoxides to the corresponding diols in the plasma, including EET/DHET, may be an indicator of the activity of hepatic sEH. 2. The conversion of EET to DHET by sEH is regioselective, and 14, 15-EET is the preferred substrate. 3. The enzymatic activity of sEH in the liver was altered in patients with MDD.	[64]
2	Blood plasma from MDD patients.	ELISA: 14, 15-DHETs (↑); 5, 6-DHET (-); 8, 9-DHET (-); 11, 12-DHET (-).		
3	Blood plasma from MDD patients.	ELISA: Total ratio of EETs/DHETs (↓); Ratio of 14, 15-EET/14, 15-DHET (↓).		
4	Human postmortem parietal cortex (Brodmann area 7) from MDD patients, BD patients, and SZ patients.	Western blot: sEH in the parietal cortex from MDD, BD, and SZ groups (↑).	1. There was a positive correlation between sEH protein in the parietal cortex and hepatic sEH. 2. The increased expression of sEH in the brain and hepatic sEH might play a role in the pathogenesis of major psychiatric disorders, 3. Brain-liver axis may be a critical part in major psychiatric disorders.	[67]
5	Human liver from MDD patients, BD patients, and SZ patients.	Western blot: sEH in the liver from MDD, BD, and SZ groups (↑).		

**Abbreviations:** Major depressive disorder (MDD); bipolar disorder (BD); schizophrenia (SZ); soluble epoxide hydrolase (sEH); epoxyeicosatrienoic acid (EET); arachidonic acid (AA); linoleic acid (LA); dihydroergotoxine (DHET); not acceptable (NA); significant increase (↑), *p* < 0.05; significant decrease (↓), *p* < 0.05; no significance (-).

**Table 3 molecules-27-07269-t003:** Association between hepatic sEH and breast cancer based on the preclinical studies.

S. No	Model/Animals/Cells Used	Administration	Outcome of the Study	Key Findings	References
1	MCF-7 cells (in vitro).	9,10- and 12,13-DiHOME (0.32–1.6 μM).	Cell proliferation (↑).	Mechanisms not involving estrogen receptor or nuclear type II binding sites.	[26]
2	MDA-MB-231 breast cancer cells (in vitro).	Stimulation with AA was performed with a solution of AA dissolved in ethanol.	p-Akt (↑); Migration (↑); Invasion (↑).	Akt/PI3K and EGFR pathways mediate migration and invasion induced by AA in MDA-MB-231 breast cancer cells.	[149]
3	MDA-MB-231 cells (in vitro).	DMEM/F12 (1:1) supplemented with 5% FBS, 10 lg/mL insulin, 0.5 lg/mL hydrocortisone, 20 ng/mL EGF, and antibiotics.	p-Akt (↑); FFAR (↑); NF-κB (↑).	LA induces migration and invasion through an EGFR/PI3K/Akt-dependent pathway in breast cancer cells.	[147]
4	Male C57BL/6 N mice (in vivo).	1. CO; 2. The conventional soybean oil diet contains 50% CO and 50% SO (SO + CO); 3. The PL + CO diet.	SO+CO group mice: hepatic AA (↑), plasma AA (↑); hepatic LA (↑), plasma LA (↑); hepatic 9, 10-DiHOME (↑), hepatic 12, 13-DiHOME (↑).	Increased LA consumption leads to increased concentrations of EpOMEs and DiHOMEs in the liver, which cause the upregulation of 9, 10-DiHOME.	[142]
5	Male C57BL/6J mice (in vivo).	A commercially available butter blend (Hiland Dairy Foods); trans-fat free margarine (Land O’Lakes); ALA-enriched butter (Sunseo Milk Butter™).	Dietary reduction in ω-6/ω-3 FA ratio is effective in reducing systemic levels of ω-6/ω-3 ratio, promoting biosynthesis of long-chain ω-3 PUFA and generation of both ALA- and EPA-derived oxylipins.	Oxidized linoleic acid metabolites induce liver mitochondrial dysfunction, apoptosis, and NLRP3 activation in mice.	[154]
6	Sf-21 cells infected with recombinant baculoviruses produce either hsEH, msEH, hmEH, or lacZ (in vitro).	2% final concentration of DMSO (vol/vol)/3 days.	The toxicity of the leukotoxins in cells expressing msEH could be reversed by the administration of the potent sEH inhibitor 4-fluorochalcone oxide.	The numerous pathologies attributed to leukotoxin and isoleukotoxin result from enzymatic activation mediated largely by the soluble epoxide hydrolase.	[73]
7	Sprague Dawley rats (in vivo).	Free fatty acids dissolved in PBS containing 5% DMSO by cardiac puncture after anesthetized by intraperitoneal injections of pentobarbital.	35 mg/kg leukotoxin diol caused immediate respiratory distress with 100% death in less than 2 h.	The leukotoxins can be metabolism by EHs in target tissues or the diols can be formed in hepatic and renal tissues with high sEH and then released into general circulation.	
8	MDA-MB-231 breast cancer cells (in vitro).	LA.	1. The formation of filopodia and lamellipodia (↑); 2. The localization of fascin (↑); 3. Migration (↑), invasion (↑), matrix metalloproteinase-9 secretion (↑).	LA induces the formation of filopodia and lamellipodia and the localization of fascin in these actin structures in MDA-MB-231 breast cancer cells. Fascin is required for migration and invasion induced by LA in MDA-MB-231 breast cancer cells.	[149]
9	MCF12A breast cancer cells (in vitro).	LA.	1. The formation of microspikes (↑); 2. The localization of fascin (↑).		

**Abbreviations:** Human soluble epoxide hydrolase (hsEH); mouse soluble epoxide hydrolase (msEH); human microsomal epoxide hydrolase (hmEH); P-galactosidase (lacZ); coconut oil diet (CO); soybean oil diet (SO); linoleic acid (LA); α-linolenic acid (ALA); oleic soybean oil, Plenish (PL); arachidonic acid (AA); not acceptable (NA); free fatty acid receptor (FFAR); dihydroxy-9Z-octadecenoic acid (DiHOME); significant increase (↑), *p* < 0.05; significant decrease (↓), *p* < 0.05; no significance (-).

## Data Availability

Not applicable.

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
