# Peer review of "Crosstalk between Depression and Breast Cancer via Hepatic Epoxide Metabolism: A Central Comorbidity Mechanism"

_molecules, 2022, doi:10.3390/molecules27217269_

Round 1

Reviewer 1 Report

The article is well written by authors and at this stage minor spelling changes are required. 

Author Response

Thank you for your comments. We would like to apologize for the linguistic problems. The whole MS was improved, and we invited an expert whose mother tongue is English to review the MS. Thank you for your valuable comments.

Reviewer 2 Report

I am thankful to the editor(s) for this opportunity for reviewing the article- “Crosstalk between Depression and Breast Cancer via Hepatic Epoxide Metabolism: A Central Co-morbidity Mechanism”.

The review is very interesting and very well drafted. This review claims the potential line for the possibility that hepatic epoxide metabolism could be a co-morbidity and etiology of breast cancer progression. This is a very novel approach for the therapy in future but may need many pre-clinical and clinical investigations. It would have been better, if the authors would have used some databases (TCGA) information to prove their claim. There are some minor errors and queries. I will request the authors to look after these points.

1.     Page No 2 line no 59…. The ref mentioned against this line is not by Stefan Schneider and Anne Moyer. Kindly correct it.

2.     Section 2, line no 107-118. A figure describing this paragraph will be helpful to the readers.

3.     There are some minor spelling errors in the manuscript. Kindly take care of it e.g., line no 169 “activity” is mis-spelled.

4.     In Table 1, the proper citations are missing. Kindly add all the citations for the various claims made.

5.     In Table 2, Sl. no 1 and 2, citation no 117 is also cited in Table-1 Sl. no 1. The ref. seems to have the information of in-vivo not of any clinical outcomes. Kindly rectify this.

6.     In section 4, line no 209-223, the paragraph is completely copied from section 2 (Page no 3, line number-105-119). It is recommended to either omit the same or change the writing pattern of the whole paragraph.

7.     It is recommended to change the title of section 6, as most of the explanations are directly related to the brain/depression rather than the breast cancer.

8.     Line no 393 to 395, kindly add some additional ref. claiming this.

9.     Line number 417 -418, it is recommended to refer some case studies (if done).

10.  The review has no direct explanation of hepatic inflammation and breast cancer although lot of explanation are provided in terms of depression and hepatic inflammation. It is suggested to discuss some direct evidence on hepatic inflammation and breast cancer. For example, authors can correlate with hepatitis B or C infection related inflammation pathways  and then link to breast cancer or breast cancer metastasis.

Author Response

I am thankful to the editor(s) for this opportunity for reviewing the article- “Crosstalk between Depression and Breast Cancer via Hepatic Epoxide Metabolism: A Central Co-morbidity Mechanism”.

The review is very interesting and very well drafted. This review claims the potential line for the possibility that hepatic epoxide metabolism could be a co-morbidity and etiology of breast cancer progression. This is a very novel approach for the therapy in future but may need many pre-clinical and clinical investigations. It would have been better, if the authors would have used some databases (TCGA) information to prove their claim. There are some minor errors and queries. I will request the authors to look after these points.

  1. Page No 2 line no 59…. The ref mentioned against this line is not by Stefan Schneider and Anne Moyer. Kindly correct it.

Re. The comments from Stefan Schneider and Anne Moyer for this citation (https://doi.org/10.1002/cncr.25318) were mis-cited. The reference was well modified. Thank you very much.

  1. Section 2, line no 107-118. A figure describing this paragraph will be helpful to the readers.

Re. The figure description was added following the relative paragraph. Thank you very much.

  1. There are some minor spelling errors in the manuscript. Kindly take care of it e.g., line no 169 “activity” is mis-spelled.

Re. The mis-spelling was modified. The MS was checked carefully. Thank you very much.

  1. In Table 1, the proper citations are missing. Kindly add all the citations for the various claims made.

Re. The citations were properly added. Thank you very much for your kind suggestion.

  1. In Table 2, Sl. no 1 and 2, citation no 117 is also cited in Table-1 Sl. no 1. The ref. seems to have the information of in-vivonot of any clinical outcomes. Kindly rectify this.

Re. The information that presented in the Table-1 is the result of animal experiment, while the information that exhibit in Table-2 is the clinical outcomes.

  1. In section 4, line no 209-223, the paragraph is completely copied from section 2 (Page no 3, line number-105-119). It is recommended to either omit the same or change the writing pattern of the whole paragraph.

Re. We would like to apologize that this paragraph is mis-copied during the format transferring. In the recent version, we deleted the paragraph in a correct way. Thank you for your kind remind.

  1. It is recommended to change the title of section 6, as most of the explanations are directly related to the brain/depression rather than the breast cancer.

Re. The title of section 6 was changed into an appropriate description. Thank you.

  1. Line no 393 to 395, kindly add some additional ref. claiming this.

Re. The additional citations were added into the MS, which were in highlight. Thank you.

  1. Line number 417 -418, it is recommended to refer some case studies (if done).

Re. The case studies were added properly. Thank you.

  1. The review has no direct explanation of hepatic inflammation and breast cancer although lot of explanation are provided in terms of depression and hepatic inflammation. It is suggested to discuss some direct evidence on hepatic inflammation and breast cancer. For example, authors can correlate with hepatitis B or C infection related inflammation pathways and then link to breast cancer or breast cancer metastasis.

Re. The direct explanation of hepatic inflammation and breast cancer was added into the MS. The relative content is in highlight (line 443-446). Thank you.

Thank you again for your patience and valuable comments, which significantly improve the quality of our MS. We’ve pay high attention to each point and carefully modified the MS. If there’re any question, please feel free to contact us.

Reviewer 3 Report

To the authors, 

I want to congratulate the authors for highlighting a perhaps overlooked concept of understanding the link between neurological disorders such as MDD and how it can potentially impact progression in breast cancer patients. This will help in developing a wholistic approach to care of breast cancer patients. However, I want to bring to your attention some concerns regarding some of the conclusions drawn in the article. The review of the manuscript is given below. 

Title: Crosstalk between Depression and Breast Cancer via Hepatic Epoxide Metabolism: A Central Co-morbidity Mechanism

Manuscript Summary

In this manuscript, Ye, Z et al. have presented a comprehensive review on the association between breast cancer and depression as a co-morbidity associated with hepatic epoxide metabolism. The authors have presented the case for how dysregulation in hepatic epoxide metabolism can affect progression of breast cancer and induction of depression by the effect of the former on inflammation, upregulation of toxic diols and disruption in lipid metabolism. The thesis presented in this review article is very interesting and perhaps highlights a novel way to think on the impact of systemic organ response to cancer development and the associated evolution of other co-morbidities. This review is a significant addition to the field and will have considerable impact on developing new ideas to understand coordination between different systemic mechanisms, which can be critical for disease progression. Given below are some of the comments in areas that need additional clarifications in the article.

Comments

11.      There are multiple areas in the article that can be further highlighted by improving on English grammar and writing.

22.  In line 55, the authors have stated that depression is an independent predictor of higher frequency, longer hospitalization, lower quality of life, and lower treatment compliance. From the sentence it is not evident what is the higher frequency referring to.

33. In line 57, the authors have stated that depression is a predictor for advanced breast cancer patients. It is not evident from the research cited that there is any etiology between depression and advanced stages of breast cancer. The authors have to clarify this point because the major objective of the article cited (citation 12) was to focus on how depression might influence clinical diagnosis of breast cancer taking into consideration psychological parameters of the patient.

44. In line 62, the authors need to clarify what is meant by relative depression.

55.  Although, the authors mention have mentioned that depression is a critical co-morbidity of breast cancer, it is not evident why only a small proportion of breast cancer patients (15%) are affected by depression. The authors have to clarify this point.

66.  Lines 105-119 and 209-223 are exact duplicates. This needs to be edited and modified.

77. Although the authors have drawn the conclusion that the depression associated molecular changes in liver can promote breast cancer by modulating the tumor microenvironment, the research presented is only suggestive of that fact. There are many critical aspects that need to be tested to address the scientific rigor of such a conclusion. Firstly, it is not evident if depression is an independent predictor of breast cancer development and progression as both MDD and breast cancer are individually a multifactorial disease and at the same time be associated with one another. Secondly, there are no evidence of any prospective studies that have been conducted to associate clinical depression with breast cancer outcome. Thirdly, breast cancer affect women and men. This then leads to the question if there are gender differences in depression onset and outcome and how that can lead to breast cancer incidence or progression. Fourthly, high fat diet or obesity, diabetes, alcohol consumption, and depression can be independent or non-independent events. Each of the cases can lead to hepatic dysregulation. It is difficult to ascertain then if depression can be significant physiological event that can induce hepatic dysregulation to induce progression of breast cancer. Addressing these points will be important to convince the audience that depression might have a significant impact of its own in inducing breast cancer development and progression. 

Author Response

To the authors, 

I want to congratulate the authors for highlighting a perhaps overlooked concept of understanding the link between neurological disorders such as MDD and how it can potentially impact progression in breast cancer patients. This will help in developing a wholistic approach to care of breast cancer patients. However, I want to bring to your attention some concerns regarding some of the conclusions drawn in the article. The review of the manuscript is given below. 

Title: Crosstalk between Depression and Breast Cancer via Hepatic Epoxide Metabolism: A Central Co-morbidity Mechanism

Manuscript Summary

In this manuscript, Ye, Z et al. have presented a comprehensive review on the association between breast cancer and depression as a co-morbidity associated with hepatic epoxide metabolism. The authors have presented the case for how dysregulation in hepatic epoxide metabolism can affect progression of breast cancer and induction of depression by the effect of the former on inflammation, upregulation of toxic diols and disruption in lipid metabolism. The thesis presented in this review article is very interesting and perhaps highlights a novel way to think on the impact of systemic organ response to cancer development and the associated evolution of othWer co-morbidities. This review is a significant addition to the field and will have considerable impact on developing new ideas to understand coordination between different systemic mechanisms, which can be critical for disease progression. Given below are some of the comments in areas that need additional clarifications in the article.

Comments

  1.    There are multiple areas in the article that can be further highlighted by improving on English grammar and writing.

Re. Thank you for your comments. We would like to apologize for the linguistic problems. The whole MS was improved, and we invited an expert whose mother tongue is English to review the MS. Thank you for your valuable comments.

  1. In line 55, the authors have stated that depression is an independent predictor of higher frequency, longer hospitalization, lower quality of life, and lower treatment compliance. From the sentence it is not evident what is the higher frequency referring to.

Re. We sincerely apologize for the mis-understanding expression. The higher frequency refers to the hospitalization. This expression was well modified. Thank you for your question, which help us to improve the inappropriate expression (line 56).

  1. In line 57, the authors have stated that depression is a predictor for advanced breast cancer patients. It is not evident from the research cited that there is any etiology between depression and advanced stages of breast cancer. The authors have to clarify this point because the major objective of the article cited (citation 12) was to focus on how depression might influence clinical diagnosis of breast cancer taking into consideration psychological parameters of the patient.

Re. We would like to express our appreciation for your suggestion. We changed our description for the relative sentence (line 58). Thank you.

  1. In line 62, the authors need to clarify what is meant by relative depression.

Re. The description is modified into an accurate way (line 62-63). Thank you.

  1. Although, the authors mention have mentioned that depression is a critical co-morbidity of breast cancer, it is not evident why only a small proportion of breast cancer patients (15%) are affected by depression. The authors have to clarify this point.

Re. The reported data that cited is a part of real world. In our previous study, the negative emotions, including depression, show significantly increase the risk for the incidence of BC (line 66-68). Thank you.

  1. Lines 105-119 and 209-223 are exact duplicates. This needs to be edited and modified.

Re. The duplicated paragraph is mis-added during the format creation. The duplicated content (line 209-223) was deleted. Thank you very much.

  1. Although the authors have drawn the conclusion that the depression associated molecular changes in liver can promote breast cancer by modulating the tumor microenvironment, the research presented is only suggestive of that fact. There are many critical aspects that need to be tested to address the scientific rigor of such a conclusion. Firstly, it is not evident if depression is an independent predictor of breast cancer development and progression as both MDD and breast cancer are individually a multifactorial disease and at the same time be associated with one another. Secondly, there are no evidence of any prospective studies that have been conducted to associate clinical depression with breast cancer outcome. Thirdly, breast cancer affect women and men. This then leads to the question if there are gender differences in depression onset and outcome and how that can lead to breast cancer incidence or progression. Fourthly, high fat diet or obesity, diabetes, alcohol consumption, and depression can be independent or non-independent events. Each of the cases can lead to hepatic dysregulation. It is difficult to ascertain then if depression can be significant physiological event that can induce hepatic dysregulation to induce progression of breast cancer. Addressing these points will be important to convince the audience that depression might have a significant impact of its own in inducing breast cancer development and progression. 

Re. Thank you for your comments. Recently, the evidence of the relationship between depression and breast cancer was supplied into the MS, which demonstrates the potential investigating value. Additionally, the relationship among high fat diet, hepatic dysregulation and breast cancer is complex, therefore we focus on the enzyme sEH. Eventually, we cannot agree more with the fact that the strict scientific evidence is still absence. And we aim to discuss the role of hepatic sEH, which might be a critical enzyme in this progress, which is also the key value of our MS.

We would like to express our high appreciation for your valuable comments and suggestion. Your comments with details significantly help us to improve the quality of our MS. If there are any questions, please feel free to contact us.

Round 2

Reviewer 3 Report

The authors have provided a comprehensive review on the crosstalk between depression and breast cancer via regulation of epoxide metabolism. The authors have made recommended changes and therefore is in acceptable format for publication.